# Alcohol’s Impact on the Gut and Liver

**DOI:** 10.3390/nu13093170

**Published:** 2021-09-11

**Authors:** Keith Pohl, Prebashan Moodley, Ashwin D. Dhanda

**Affiliations:** 1South West Liver Unit, University Hospitals Plymouth NHS Trust, Plymouth PL6 8DH, UK; k.pohl@nhs.net (K.P.); p.moodley@nhs.net (P.M.); 2Hepatology Research Group, Faculty of Health, University of Plymouth, Plymouth PL4 8AA, UK

**Keywords:** alcohol, gut, liver, cirrhosis, hepatocellular carcinoma, microbiome

## Abstract

Alcohol is inextricably linked with the digestive system. It is absorbed through the gut and metabolised by hepatocytes within the liver. Excessive alcohol use results in alterations to the gut microbiome and gut epithelial integrity. It contributes to important micronutrient deficiencies including short-chain fatty acids and trace elements that can influence immune function and lead to liver damage. In some people, long-term alcohol misuse results in liver disease progressing from fatty liver to cirrhosis and hepatocellular carcinoma, and results in over half of all deaths from chronic liver disease, over half a million globally per year. In this review, we will describe the effect of alcohol on the gut, the gut microbiome and liver function and structure, with a specific focus on micronutrients and areas for future research.

## 1. Introduction

Alcohol (ethanol) is a small water-soluble molecule that enters the blood stream via the stomach and proximal small intestine and is then distributed throughout the body. It first enters the portal vein, which drains directly into the liver, where the greatest exposure to alcohol occurs. The liver eliminates the majority of alcohol (90%), while 2–5% is excreted unchanged in urine, sweat and breath [1].

Alcohol consumption is engrained in many cultures, making alcohol the most commonly used drug worldwide. However, it is not without risk. Globally, alcohol is the seventh leading cause of death and disability-associated life years (DALYs) lost, and it caused 2.8 million deaths in 2016 [2]. In that year, among adults less than 50 years old, alcohol was the leading cause of death and DALYs lost, responsible for 3.8% and 12.2% of female and male deaths, respectively [2]. Alcohol is causally implicated in over 200 conditions, including cancers of the digestive tract and liver [3]. However, a large proportion of the global burden is due to alcohol-related liver disease, accounting for 27% of all deaths from chronic liver disease, and alcohol-related hepatocellular carcinoma, together responsible for over half a million deaths annually [4,5].

Here, we review the effect of alcohol on the gut and liver, focusing on its interaction with micronutrients.

## 2. Alcohol and the Gut

The pathological effects of alcohol on the digestive system hinge in part on the gut–liver axis. This bi-directional relationship facilitated by the enterohepatic circulation involves the transportation of digestive and bacterial products from the gut to the liver, and the return of bile, antibodies and cytokines to the gut [6]. Alcohol ingestion in both chronic and ‘binge’ settings has been shown to alter this axis through the disruption of gut microbial composition, the metabolome and the gut epithelial barrier. These disturbances ultimately have a knock-on effect on nutrient absorption [7,8].

### 2.1. The Effect of Alcohol on Microbial Composition and Gut Barrier Function

The human gut microbiome describes a complex community of bacteria, viruses, fungi and archaea, which varies both with environmental factors (such as diet and drugs) and age, but less so with host genetics [9,10,11,12]. Disruption of the microbiome (dysbiosis) has been linked with a wide range of conditions including diabetes, obesity, cardiovascular disease, inflammatory bowel disease and liver cirrhosis; however, it is unclear whether this is a cause or effect relationship [13,14]. Several studies have investigated the effect of alcohol consumption in both animal and human models, and have consistently shown that alcohol consumption is linked with the development of dysbiosis [13,15]. In brief, alcohol has been shown to increase the relative abundance of *Proteobacteria*, *Enterobacteriacea* and *Streptococcus* and decrease the abundance of *Bacteroides*, *Akkermansia* and *Faecalibacterium* [15]. The aetiological mechanism of this dysbiosis is not fully understood; however, it is likely that this is multifaceted, including alcohol-induced oxidative stress (which is poorly tolerated by obligate anaerobes such as *Bacteroides*) and the downregulation of antibacterial peptides such as α-defensins by alcohol [16,17].

Alcohol-induced dysbiosis contributes to the development of both acute (e.g., alcoholic hepatitis) and chronic (e.g., alcohol-related cirrhosis) liver diseases through its pathological effect on gut integrity. The intestinal mucous barrier has an essential role in the immune function of the gut, and its disruption leads to these disease states. In this barrier, neighbouring enterocytes are bound together by the apical ‘tight junction’ proteins claudins, occludin and zona occludens, preventing the unwanted translocation of luminal contents such as pathogen-associated molecular particles (PAMPs) and bacterial endotoxins into the portal circulation [18]. Dysbiosis induced by alcohol consumption has been linked to the disruption of these tight junctions. As a consequence, the subsequent immune dysfunction and increase in circulating pro-inflammatory cytokines such as tumour necrosis factor (TNF)-α and interleukin (IL)-1β further disrupts the gut barrier [18,19].

### 2.2. The Effect of Alcohol on the Metabolome

Alcohol-related dysbiosis inevitably affects the gut metabolome, and dramatic alterations in short-chain fatty acids (SCFAs), amino acids and bile acids have been documented.

The role of SCFAs in the maintenance of tight junctions is becoming increasingly apparent. SCFAs are fatty acids with fewer than six carbon atoms, and are the product of the anaerobic fermentation of indigestible dietary fibres by the gut microbiota [20]. Analysis of the faecal metabolome in humans with alcohol use disorders revealed a reduction in SCFAs, which is likely to be due in part to dysbiosis that negatively affects SCFA-producing bacteria such as *Faecalibacterium* [15,21]. Several murine models have reliably shown that supplementation with SCFAs in either the form of a high fibre diet, probiotic or dietary modification enhances gut epithelial integrity and reduces liver injury in alcoholic models, and work in this area is ongoing [22,23,24].

The gut metabolome also plays an important part in the metabolism and absorption of essential and non-essential amino acids, which appear to be altered by alcohol ingestion. Less work has been conducted to investigate this effect. However, several studies have identified that alcohol consumption lowers the concentration of almost all amino acids in the gut lumen [12,14,25]. Both essential, dietary-obtained amino acids (e.g., lysine) and non-essential amino acids (e.g., glutamic acid) are affected. It is postulated that this is a result of a disturbed microbial–host co-metabolism as a result of dysbiosis [14]. Although luminal amino acid concentrations fall with alcohol consumption, serum levels of some, such as tyrosine and phenylalanine, rise, suggesting an altered metabolic and absorption profile of the dysbiotic microbiome [14,26]. This metabolic imbalance may play a role in the generation of increased levels of reactive oxygen species (ROS) and toxic intermediates.

Bile acids have been shown to be altered in both the serum and luminal contents of humans and rats consuming alcohol [12,14,27]. Primary (synthesised by the liver) and secondary (from bacterial metabolism) bile acids perform a variety of functions predominantly in the small bowel and have crucial roles in lipid absorption, cholesterol homeostasis as well as hormonal actions through their steroid structure. In a healthy entero-hepatic circulation, primary bile acids are conjugated with either taurine or glycine to form bile salts that are secreted into the intestinal lumen. The intestinal microbiota then metabolises these to secondary bile acids, removing the taurine/glycine groups before recycling them back to the liver. Alcohol consumption appears to disrupt this by increasing the proportion of secondary bile acids and the total concentration of bile acids, as well as increasing the proportion conjugated with glycine instead of taurine [12,14]. It is felt that this is caused by dysbiosis decreasing the bioavailability of taurine and an increased rate of entero-hepatic cycling [12,14]. The consequence of this disruption is not fully understood; however, it is likely that the glycine-conjugated acids that are more prevalent during alcohol consumption are relatively more toxic, and the increased synthesis of bile acids despite high luminal concentrations contributes to hepatic steatosis [12].

### 2.3. The Effect of Alcohol Consumption on Nutritional Status

Chronic alcohol ingestion reduces nutrient absorption and contributes to malnutrition [28]. Alterations in intestinal permeability, bile acid profiles and the microbiome all contribute to this, and in addition, the toxic metabolites and ROS released during alcohol metabolism cause structural damage to the intestine. In particular, chronic alcohol use has been shown to cause cell death, mucosal erosions and the loss of epithelium at the villi tips [29]. The consequences of this are variable deficiencies in vitamins A, B1 (thiamine), B2 (riboflavin), B6 (pyridoxine), C, D, E and K as well as folate, calcium, magnesium, phosphate, iron and the trace elements zinc and selenium [28,30]. It is important that all patients with chronic alcohol use disorders undergo a full nutritional assessment as these deficiencies vary between individuals, with iron as an example that can be either deficient or found in excess. Alongside the mechanisms described above, heavy alcohol users obtain up to 50 percent of their daily caloric intake from nutritionally deplete alcoholic drinks [8]. Furthermore, it should be noted that alongside the symptomatic effects of chronic alcohol misuse (e.g., vomiting, anorexia and abdominal pain), social factors in this group such as poverty and access to a nutritionally ‘complete’ diet may also contribute to malnutrition [31].

The effect of alcohol on the gut is summarised in Figure 1.

## 3. Alcohol and the Liver

### 3.1. Alcohol Metabolism

The metabolism of alcohol in the liver is key to understanding its role in the pathogenesis of alcohol-related liver disease. Alcohol is primarily metabolised in hepatocytes by alcohol dehydrogenase to acetaldehyde and then to acetate by aldehyde dehydrogenase. Acetate is converted to water and carbon dioxide mainly in peripheral tissue, which is easily excreted. A minority of alcohol is metabolised by the mitochondrial enzyme oxidation system (MEOS), through the action of the cytochrome P450 (CYP) enzyme CYP2E1, to acetaldehyde with the generation of ROS. A third minor pathway of alcohol metabolism to acetaldehyde is by the action of catalase and the conversion of H_2_O_2_ to H_2_O.

It is the generation of acetaldehyde, a highly reactive protein, which contributes to liver damage. It binds to lipids, proteins and DNA to form potentially immunogenic adducts [32]. These adducts can generate an adaptive immune response leading to hepatocellular damage and inflammation [33]. Structural mitochondrial alteration can lead to functional impairment including decreased ATP generation, the production of ROS and decreased activity of acetaldehyde dehydrogenase. Acetaldehyde is also a key metabolite in the progression of liver fibrosis. It can promote the synthesis of collagen I in hepatic stellate cells (HSCs), and acetaldehyde adducts stimulate the release of inflammatory cytokines and chemokines [33].

The alcohol dehydrogenase pathway is efficient in metabolising alcohol in small quantities, but in chronic alcohol exposure, the pathway becomes saturated and there is significant induction of CYP2E1 [32]. The switch to the CYP pathway results in the generation of ROS, leading to oxidative stress. ROS bind to proteins, changing their structural and functional properties, and may act as neoantigens. ROS can also bind directly to DNA, causing damage, or lead to lipid peroxidation products such as 4-hydroxynonenal (4-HNE) and malondialdehyde (MDA) that generate highly carcinogenic DNA adducts (Figure 2) [34]. In addition, in chronic heavy alcohol ingestion, the antioxidant clearing system of the liver is impaired because of an acetaldehyde-mediated decrease in glutathione. The outcome of oxidative stress is the induction of hepatocyte apoptosis and necrosis [35].

### 3.2. Alcohol-Related Steatosis

Steatosis, characterised by the accumulation of fat (triglycerides, phospholipids and cholesterol esters) in hepatocytes, is the earliest response of the liver to chronic alcohol use and is almost universal in chronic heavy drinkers [36]. Although it is fully reversible upon a reduction in alcohol use, its presence is associated with the progression of alcohol-related liver disease, with a recent meta-analysis finding an annual progression rate to cirrhosis of 3% [37]. It is likely that hepatic steatosis increases the risk of liver inflammation (steatohepatitis), fibrosis and cirrhosis through greater lipid peroxidation and oxidative stress. However, progression only occurs in up to 20% and is not only influenced by the amount of alcohol but also other factors, including gender, co-existing liver disease, smoking and genetics [38].

Chronic alcohol ingestion leads to hepatic steatosis via increased hepatic lipogenesis and decreased hepatic lipolysis. Alcohol elevates the ratio of reduced NAD/oxidised NAD in hepatocytes, which interferes with mitochondrial beta oxidation of fatty acids, leading to their accumulation in hepatocytes [39]. Chronic alcohol use induces hepatic expression of sterol regulatory element binding protein-1c (SREBP1c), a transcription factor that stimulates the expression of lipogenic genes, resulting in increased fatty acid synthesis [40]. Alcohol also downregulates inhibitors of SREBP1c expression such as AMP-activated protein kinase (AMPK), Sirtuin-1, adiponectin and signal transducer and activator of transcription 3 (STAT3) [41]. Conversely, chronic alcohol use enhances adipose tissue breakdown and lipolysis, releasing free fatty acids, which are esterified in hepatocytes into triglycerides [42].

Alcohol inactivates peroxisome proliferator-activated-receptor (PPAR)-α, a nuclear hormone receptor that upregulates the expression of many genes involved in free fatty acid transport and oxidation. Acetaldehyde directly inhibits transcriptional activation activity and DNA binding of PPAR-α [43]. Alcohol also indirectly inhibits PPAR-α via CYP2E1-derived oxidative stress, adenosine, the downregulation of adiponectin and zinc deficiency (a common state in patients with alcohol-related liver disease) [30,44]. The inactivation of PPAR-α results in reduced hepatic lipolysis (Figure 3).

### 3.3. Alcoholic Steatohepatitis

Hepatic inflammation strongly influences the development of fibrosis, cirrhosis and ultimately hepatocellular carcinoma. The alcohol-induced leaky gut (as described above) leads to the delivery of PAMPs to the liver. Together with damage-associated molecular patterns, released from damaged cells, PAMPs activate innate receptors (Toll-like receptors (TLRs) and NOD-like receptors (NLRs)) on monocytes, macrophages, Kupffer cells and hepatic parenchymal cells. Signalling through these receptors leads to increased transcription of pro-inflammatory transcription factors including nuclear-factor κB (NFκB) and the production of pro-inflammatory chemokines and cytokines (reviewed in detail in ref [45]). The net effect is the influx of monocytes, neutrophils and T cells, the release of soluble mediators that cause cell death and hepatic stellate cell (HSC) activation (Figure 4). In addition to an activated pro-inflammatory immune response to alcohol, patients with alcoholic hepatitis have evidence of immune dysfunction. The activation of monocytes by gut-derived PAMPs leads to T cell exhaustion with reduced numbers of anti-inflammatory IL-10 producing T cells and functionally impaired monocytes and neutrophils [46,47].

Hepatocyte cell death occurs through several mechanisms including apoptosis, pyroptosis, necrosis and necroptosis [48]. Apoptosis is induced by direct alcohol-mediated hepatotoxicity, the induction of oxidative stress, the inhibition of survival genes (*C-met*) and the induction of pro-apoptotic signalling molecules (TNF-α and Fas ligand) [49]. Necrosis, cell swelling and membrane rupture can also occur via a programmed pathway known as necroptosis, while pyroptosis is a programmed cell death dependent on caspase-1. The mode of cell death is likely to be influenced by the disease state, with apoptosis predominating in early alcohol-related liver disease but inflammasome activation driving pyroptosis and propagating liver injury in alcoholic hepatitis [50].

MicroRNAs (miRNAs) are small non-coding RNAs that have a role in the post-transcriptional regulation of their target genes. Two key miRNAs are differentially expressed in patients with alcohol-related liver disease. miRNA-155, a key regulator of inflammation, is increased in the liver and circulation in mouse models of alcohol-related liver disease [51]. Chronic alcohol consumption increases the expression of miRNA-155 in Kupffer cells, which contributes to increased LPS-triggered TNF production [52]. miR-181b-3p, a negative regulator of TLR4 signalling in Kupffer cells, is downregulated in patients with alcohol-related liver disease [53].

### 3.4. Alcohol-Induced Fibrosis and Cirrhosis

Fibrosis is the liver’s wound healing response to a damaging stimulus, reversible on removal of the stimulus. In the presence of heavy long-term alcohol consumption, there is chronic inflammation and fibrogenesis causing the deposition of broad bands of fibrous tissue, distorting the liver architecture and altering hepatic blood flow, leading to portal hypertension and its associated complications. Extracellular matrix deposition by activated HSCs is the key event in the development and progression of liver fibrosis. Other cells (portal fibroblasts and myofibroblasts) contribute to a smaller extent [54]. HSCs are activated both by inflammatory cytokines and directly by alcohol and its metabolites and ROS. Activated HSCs perpetuate the inflammatory response by the secretion of chemokines and the expression of adhesion molecules that attract and stimulate circulating immune cells, which in turn activate quiescent HSCs [55].

### 3.5. Hepatocellular Carcinoma

Cirrhosis is a precancerous state increasing the risk of primary liver cancer, the commonest being hepatocellular carcinoma (HCC). Globally, around 30% of HCCs are due to alcohol. Alcohol itself is a carcinogen and in the context of HCC plays specific roles in its development through ROS-induced damage, inflammatory mechanisms and its reactive metabolite, acetaldehyde.

In heavy alcohol drinkers, increased activity of the CYP pathway generates ROS, leading to DNA damage that causes cell cycle arrest and apoptosis and disrupts gene function, increasing carcinogenesis and propagation [56]. The activation of inflammatory pathways in patients with alcohol-related liver disease is associated with increased cancer risk, although mechanisms have not been fully elucidated but are likely to involve pro-inflammatory cytokine promotion of ROS accumulation [57,58]. Cytokine generation is also associated with the upregulation of the angiogenesis and metastasis development [59]. Additionally, alcohol suppresses the anti-tumour response of CD8+ T cells [60].

Acetaldehyde is highly reactive and forms adducts with DNA and proteins that cause mitochondrial damage and disruption of DNA repair mechanisms. Increased levels of acetaldehyde found in people with genetic variations that confer altered activity of alcohol dehydrogenase and aldehyde dehydrogenase are associated with a higher risk of HCC in heavy drinkers [61].

## 4. Research Priorities and Future Perspectives

The human body is able to metabolise and eliminate small volumes of alcohol without long-term sequelae. However, excessive alcohol use leads to alterations in the gut microbiome, metabolome, epithelial integrity and immune signalling culminating in progressive liver disease. Our recent improved understanding of these changes has identified potential new therapies to delay or reverse liver disease. Here, we suggest areas in need of further research and potential strategies for therapy.

### 4.1. Microbiome

Alcohol’s effect on the microbiome is highly heterogeneous with multiple genera up- or downregulated, making a single probiotic treatment challenging to identify. Probiotics marginally improve liver function tests in patients with liver disease [62], but few trials in patients with alcohol-related liver disease have been conducted and they have failed to demonstrate any clinical benefit in patients with cirrhosis [63]. However, combination therapy to restore gut dysbiosis may be a more successful strategy.

Faecal microbiota transplant has recently been piloted as a treatment for patients with alcohol use disorder [64]. Healthy donor stool transplant by nasogastric injection was associated with a partial improvement in bacterial diversity and reversal in dysbiosis, an improvement in gut SCFA production and reduced severity of alcohol use disorder [64]. Although not designed to treat or prevent cirrhosis, this therapy may prove beneficial in this patient population.

### 4.2. Short-Chain Fatty Acids

SCFAs are a product of the bacterial digestion of dietary fibre and are essential to maintain gut epithelial integrity. Strategies to increase intestinal SCFA may reduce gut permeability and exposure of the liver to gut-derived toxins, thus preventing the progression of liver disease. Studies of SCFA treatment have yet to be conducted in patients with alcohol-related liver disease. Delivery of the SCFA butyrate by enema reduced gut oxidative stress and inflammation in patients with inflammatory bowel disease [65]. Indirect methods to increase SCFAs such as by faecal microbiota transplant or augmenting SCFA-producing bacteria with specific probiotics (e.g., *Clostridium butyricum*) may also be beneficial. In a randomised trial, *C. butyricum* in combination with *Bifidobacterium infantis* reduced symptoms of minimal hepatic encephalopathy in patients with hepatitis B cirrhosis as well as measures of gut permeability [66]. However, supplementation with other probiotics (*Bifidobacterium, Lactobacillus* and *Lactococcus* genera) did not improve gut barrier function [67], suggesting the importance of targeting SCFA-producing species.

Interestingly, aerobic fitness also influences the microbiome and response to dietary fat or alcohol. An animal study demonstrated that high-aerobically fit rats fed a high fat diet were protected from steatosis [68]. Conversely, low-aerobically fit rats had reduced SCFA-producing bacteria and altered microbiome metabolism of carbohydrates and energy, leading to hepatic steatosis [69]. A combined study enhancing both aerobic fitness and SCFA-producing bacteria in patients with alcohol-related liver disease may yield beneficial results.

### 4.3. Dietary Manipulation

In a cohort study comparing the gut microbiome in patients with cirrhosis from the USA and Turkey, several dietary factors were associated with a reduced risk of progression mediated by increased microbial diversity [70]. Fermented milk, vegetables, cereals, coffee and tea were associated with greater diversity, while carbonated drinks, pork and poultry were associated with lower diversity. Increased intake of insoluble fibre may also increase the bacterial production of SCFAs. Trials have been conducted in patients with hepatic encephalopathy demonstrating the safety of a high protein and high fibre diet [71,72], but none has yet been conducted evaluating the effects of long-term clinical outcomes. Further carefully designed trials of dietary manipulation are required in patients with alcohol-related liver disease.

### 4.4. Micronutrient Supplementation

Patients with alcohol-related liver disease are deficient in micronutrients including zinc and selenium [30]. Both these trace elements are essential for cellular and immune function [73,74] and their deficiency is associated with more advanced disease [75] and mortality from alcoholic hepatitis [30]. Supplementation may improve clinical outcomes through the modulation of immune function, but existing studies are too small and heterogeneous to demonstrate improved survival [76]. An ongoing trial of zinc supplementation in patients with alcohol-related cirrhosis (NCT02072746) demonstrated an improvement in liver inflammation in an interim analysis [77].

Vitamin deficiencies, especially of vitamin C and D, in patients with alcohol-related liver disease are common. Vitamin supplementation may modulate the gut microbiome resulting in reduced dysbiosis, which may reduce gut permeability and liver inflammation; studies in healthy individuals have demonstrated beneficial shifts in microbial genera with vitamin C [78] and D [79]. Although studies of antioxidant and vitamin supplementation in patients with severe alcoholic hepatitis did not demonstrate a short-term survival benefit [80], trials of long-term vitamin supplementation in patients with less acute presentations of alcohol-related liver disease are lacking.

### 4.5. Immune Dysregulation

Patients with alcohol-related liver disease have simultaneous immune activation and exhaustion [19]. Interventions targeting inflammation broadly with corticosteroids [81] or specific pro-inflammatory cytokines such as TNF-α [82] have been investigated in severe forms of alcohol-related liver disease with alcoholic hepatitis. However, increased mortality from infective complications was noted, demonstrating the delicate balance of the immune response in these patients [83]. Strategies to target immune exhaustion may promote liver repair mechanisms at the same time as maintaining defence against pathogens. Checkpoint inhibitors such as anti-PD-1 monoclonal antibodies improve the host immune response and have been licenced for the treatment of cancers. Such treatment may reduce PAMP-induced CD8+ T cell exhaustion [47] and improve the healing response in alcohol-related liver disease.

### 4.6. Repurposing of Existing Therapies

Clinical trials of commonly used drugs such as metformin, pioglitazone and statins have shown benefit on liver biochemistry or histological changes in patients with non-alcoholic steatohepatitis [84,85,86]. Benefits are mainly mediated by a reduction in insulin resistance, a mechanism that is not a strong contributor to alcohol-related liver disease. However, these drugs also alter the gut microbiome [87]; metformin, for example, increases the abundance of SCFA-producing bacteria [88]. Further trials of these agents are required to evaluate their benefit in the setting of alcohol-related liver disease.

## 5. Conclusions

In conclusion, alcohol has wide-ranging effects on the gut and liver, altering the gut microbiome, barrier function and immune function resulting in liver inflammation, fibrosis and cirrhosis. Trials of new interventional strategies to target gut dysbiosis and immune dysfunction are now required.

## Figures and Tables

**Figure 1 nutrients-13-03170-f001:**
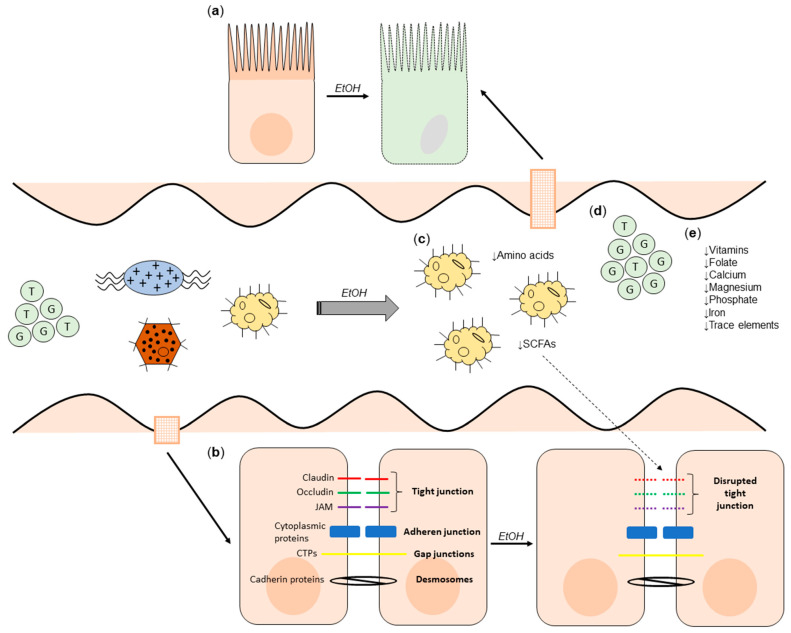
The effect of alcohol on the gut. (**a**) The histological effects of alcohol on the gut mucosa (cell death, mucosal erosions and loss of epithelium at villi tips). (**b**) Alcohol-induced disruption of tight junctions, exacerbated by reduced luminal SCFA concentrations. (**c**) Alcohol-induced dysbiosis leading to reduced SCFA and amino acid concentrations. (**d**) Increased concentration of secondary bile acids, and increased proportion conjugated with glycine. (**e**) Nutrient deficiencies as a consequence of (**a**–**d**). CTP: Connexin transmembrane proteins; JAM: Junctional adhesion molecule; EtOH: alcohol.

**Figure 2 nutrients-13-03170-f002:**
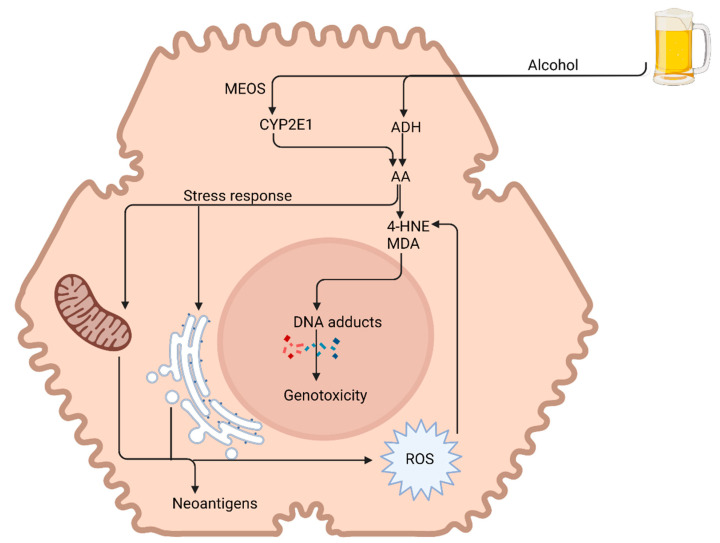
Alcohol-induced liver injury. Acetaldehyde (AA) is responsible for the majority of the toxic effects of alcohol on the liver. Acetaldehyde is extremely lipophilic, leading to the formation of acetaldehyde adducts—malondialdehyde (MDA) and 4-hydroxynonenal (4-HNE). This along with reactive oxygen species (ROS) leads to DNA damage and genotoxicity. Acetaldehyde also induces functional and structural alterations in various cell organelles (e.g., mitochondria and endoplasmic reticulum). MEOS: mitochondrial enzyme oxidation system; ADH: alcohol dehydrogenase. Image created at biorender.com (accessed on 20 August 2021).

**Figure 3 nutrients-13-03170-f003:**
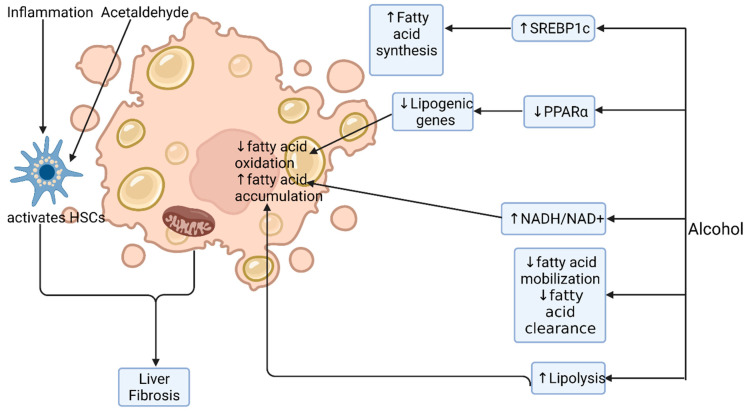
Alcohol-induced steatosis. Alcohol induces hepatic steatosis by multiple mechanisms. It alters the redox ratio in the cell (NADH/NAD+), thereby inhibiting fatty acid oxidation and promoting its accumulation. It increases transcription factor SREBP1c, which leads to increased fatty acid synthesis and deposition. Alcohol inactivates PPARα, a nuclear hormone receptor that regulates many of the genes involved in fatty acid transport and oxidation. Alcohol has a direct inhibitory effect on fatty acid clearance and mobilisation. ↑: increased; ↓: decreased; HSC: hepatic stellate cell. Image created at biorender.com (accessed on 20 August 2021).

**Figure 4 nutrients-13-03170-f004:**
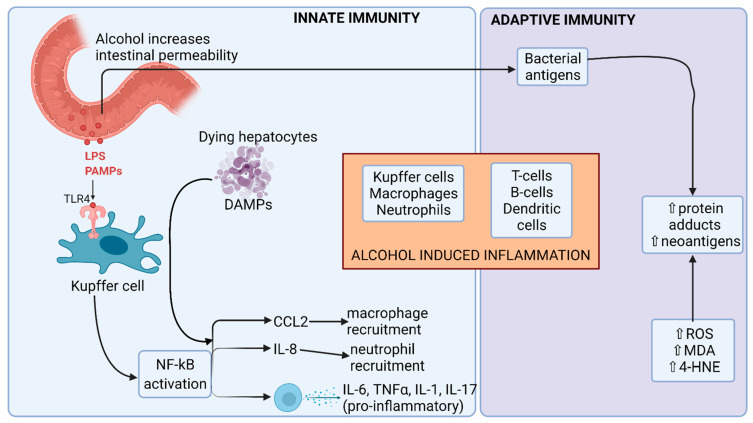
Alcohol-induced inflammation. Alcohol exerts its effects on both the innate and adaptive immunity. Alcohol not only induces enteric dysbiosis, but also increases intestinal permeability. Pathogen-associated molecular patterns (PAMPs) such as lipopolysaccharide (LPS) interact with TLR4 receptor on Kupffer cells and produce proinflammatory cytokines and chemokines via the NF-κB pathway, leading to liver inflammation. Acetaldehyde induces structural changes in various proteins and generates neoantigens, which elicit an adaptive immune response and contribute to liver inflammation. CCL2: C-C motif chemokine ligand 2; DAMPs: damage-associated molecular patterns; 4-HNE: 4-hydroxynonenal; IL: interleukin; MDA: malondialdehyde; NF-κB: nuclear factor kappa B; ROS: reactive oxygen species; TLR4: toll-like receptor 4; TNFα: tumor necrosis factor alpha; ↑: increased; ↓: decreased. Image created at biorender.com (accessed on 20 August 2021).

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
