# Peer review of "Alcohol’s Impact on the Gut and Liver"

_nutrients, 2021, doi:10.3390/nu13093170_

Round 1

Reviewer 1 Report

Comments:

The review article “Alcohol’s impact on the gut and liver” (Manuscript ID# nutrients-1371312)” provides the adverse effect of alcohol use on gut microbiota and liver injury. In addition, linked dysbiosis with liver diseases development.  The review is useful as proving basic understanding of alcohol metabolism and impact on gut-liver axis.  There are only few minor issues with the current review article:

Comment 1: “4.4. Micronutrient supplementation” section: Few lines/studies can be added related with role of vitamins in gut microbiota maintenance.

Comment 2:  “4. Research Priorities and Future Perspectives” section: one new section dealing with use of current drugs such as metformin and their effects on gut microbiota and alcoholic liver disease can be added along with need of safer drug development.

Comment 3:  Line 31-32: “proximal ………… distributed throughout the water in the body.” Sentence need to be modified.

Author Response

We thank the reviewer for their supportive comments.

Comment 1: “4.4. Micronutrient supplementation” section: Few lines/studies can be added related with role of vitamins in gut microbiota maintenance.

Response: An additional paragraph has been added to section 4.4 to describe the role of vitamin supplementation and its effect on the microbiome.

Comment 2:  “4. Research Priorities and Future Perspectives” section: one new section dealing with use of current drugs such as metformin and their effects on gut microbiota and alcoholic liver disease can be added along with need of safer drug development.

Response: A new section entitled ‘4.6. Repurposing of existing therapies’ has been added to describe the potential role of currently used drugs for the treatment of alcohol-related liver disease.

Comment 3:  Line 31-32: “proximal ………… distributed throughout the water in the body.” Sentence need to be modified.

Response: This sentence has been altered for clarity.

Reviewer 2 Report

Very interesting review. Please see minor suggestions that the authors may want to consider.

Author Response

Comment 1: Shouldn’t this be the 1st paragraph?

Response: The 1st and 2nd paragraphs have been swapped.

Comment 2: Page 2 line 45, ref please

Response: Two references have been added to support this statement.

Comment 3: Page 6 line 203. PMID: 33232792

Response: Thank you for referring us to this paper. The tailgate study examined the effect of excess energy intake in a small sample of healthy males in the form of alcohol and food. The study suggests that acute excess energy intake may have different effects on de novo lipogenesis and intrahepatic triacylglycerols depending on whether it is predominantly alcohol or carbohydrate. Although an interesting study, it is not strictly relevant to the current review, which concentrates on data describing the effects of alcohol (mainly chronic use) but not nutrition on the gut and liver.

Comment 4: Hepatic inflammation is a prerequisite for the development of fibrosis, cirrhosis and ultimately hepatocellular carcinoma – not necessarily! Is there a reference?

Response: Thank you for pointing this out. This sentence has been altered as we acknowledge fibrosis and cirrhosis can develop without the absolute requirement for inflammation.

Comment 5: Page 8 section 4. Are these recommendations?

Response: Section 4 is not a list of recommendations but suggests areas of future research and potential strategies for therapy based on our review of the literature. A sentence has been added to clarify this.

Comment 6: Page 9 section 4.2. PMID 27288420

Response: Thank you for highlighting this interesting and relevant article. A further paragraph describing the study and its implications has been added to section 4.2.

Comment 7: Very interesting article to read. Did the authors think about bariatric surgery, alcohol use, and liver disease? It’s a bit stretch for this paper but can be another interesting review.

Response: Thank you for your supportive comment. We have not been able to include the effect on bariatric surgery on the microbiome and how this might alter alcohol’s effect on the gut and liver as it is not within the remit of this review. We agree it is a very interesting area that requires a separate detailed review.